# PROTECTING DNNS FROM THEFT USING AN ENSEMBLE OF DIVERSE MODELS

**Sanjay Kariyappa**
Georgia Institute of Technology
Atlanta GA, USA
`sanjaykariyappa@gatech.edu`

**Atul Prakash**
University of Michigan
Ann Arbor MI, USA
`aprakash@umich.edu`

**Moinuddin K Qureshi**
Georgia Institute of Technology
Atlanta GA, USA
`moin@gatech.edu`

## ABSTRACT

Several recent works have demonstrated highly effective model stealing (MS) attacks on Deep Neural Networks (DNNs) in black-box settings, even when the training data is unavailable. These attacks typically use some form of Out of Distribution (OOD) data to query the target model and use the predictions obtained to train a clone model. Such a clone model learns to approximate the decision boundary of the target model, achieving high accuracy on in-distribution examples. We propose *Ensemble of Diverse Models (EDM)* to defend against such MS attacks. EDM is made up of models that are trained to produce dissimilar predictions for OOD inputs. By using a different member of the ensemble to service different queries, our defense produces predictions that are highly discontinuous in the input space for the adversary's OOD queries. Such discontinuities cause the clone model trained on these predictions to have poor generalization on in-distribution examples. Our evaluations on several image classification tasks demonstrate that EDM defense can severely degrade the accuracy of clone models (up to 39.7%). Our defense has minimal impact on the target accuracy, negligible computational costs during inference, and is compatible with existing defenses for MS attacks.

## 1 INTRODUCTION

MS attacks allow an adversary with black-box access to the predictions of the target model to copy its functionality and create a high-accuracy clone model, posing a threat to the confidentiality of proprietary DNNs. Such attacks also open the door to a wide range of security vulnerabilities including adversarial attacks (Goodfellow et al., 2014) that cause misclassification, membership-inference attacks (Shokri et al., 2017) that leak membership, and model-inversion attacks (Fredrikson et al., 2015) that reveal the data used to train the model. MS is carried out using the principle of Knowledge distillation (KD), wherein the adversary uses a dataset $\mathcal{D}$ to query the target model. The predictions of the target on $\mathcal{D}$ are then used to train a clone model that replicates the target model's functionality. Since access to training data is limited in most real-world settings, attacks typically use some form of OOD data to perform KD. Clone models trained in this way closely approximate the decision boundaries of the target model, achieving high-accuracy on in-distribution examples.

The goal of this paper is to defend against MS attacks by creating a target model that is inherently hard to steal using Knowledge Distillation with OOD data. Our key observation is the existing MS attacks (Orekondy et al., 2019a; Papernot et al., 2017; Juuti et al., 2019) implicitly assume that the target model produces continuous predictions in the input space. We hypothesise that making the predictions of the target model discontinuous makes MS attacks harder to carry out. To this end, we propose *Ensemble of Diverse Models* (EDM) to defend against MS attacks. The models in EDM are trained using a novel *diversity loss* to produce dissimilar predictions on OOD data. Each input query to EDM is serviced by a single model that is selected from the ensemble using an input-based

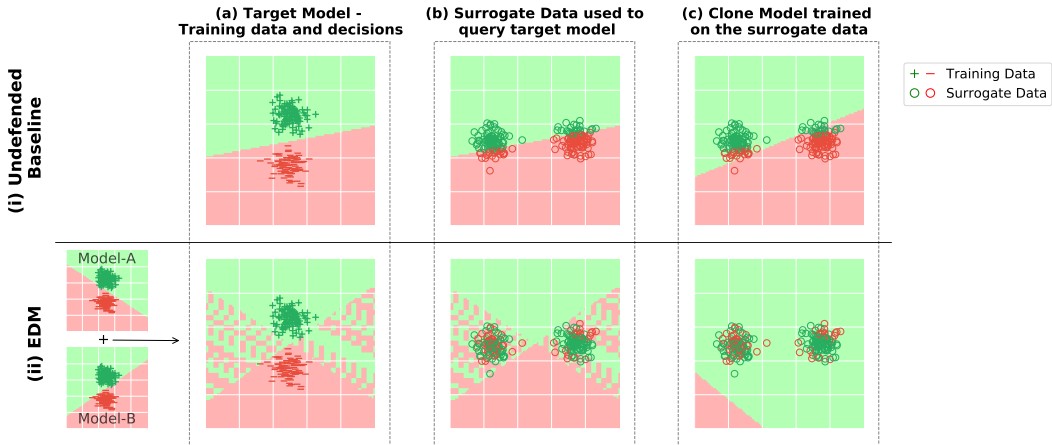

Figure 1: Toy experiment comparing the efficacy of MS attacks between (i) Undefended Baseline and (ii) EDM targets. (a) Predictions and training data of the target model (b) OOD surrogate data used by the adversary to query the target (c) Clone model trained using the predictions of the target on the surrogate data. The discontinuous predictions produced by EDM make MS attacks less effective compared to an undefended baseline. (Best viewed in color)

hashing function. We develop a DNN-based perceptual hashing algorithm for this purpose, which is invariant to simple transformations of the input and prevents adaptive attacks. Since different models in the ensemble are used to service different queries and the models are trained to produce dissimilar predictions, the adversary obtains predictions that are highly discontinuous in the input space. The clone model, when trained on these predictions, tries to approximate the complex discontinuous decision boundary of the EDM. Our empirical evaluations show that the resulting clone model generalizes poorly on in-distribution data, degrading clone accuracy and reducing the efficacy of MS attacks. In contrast to existing defenses that rely on perturbing output predictions, our defense does not require modifying the output and instead uses a diverse ensemble to produce discontinuous predictions that are inherently hard to steal.

We illustrate the working of our defense by comparing the results of MS attacks on two target models– (i) Undefended baseline (ii) EDM – trained on a toy binary-classification problem shown in Fig. 1. The training data and predictions of the target model under attack are shown in Fig. 1a. We use a set of OOD points as shown in Fig. 1b to query the target model and obtain its predictions. The predictions of the target model on the OOD data is finally used to train the clone model to replicate the functionality of the target as shown in Fig. 1c.

For the *undefended baseline* target, which uses a simple linear model to produce continuous predictions, the clone model obtained by the attack closely approximates the decision boundary of the target model, achieving good classification accuracy on in-distribution data. The EDM target consists of two diverse models (Model-A and Model-B) that produce dissimilar predictions on the OOD data. EDM uses an input-based hash function to select the model that services the input query. As a result, the target model produces highly discontinuous decisions on OOD data (Fig. 1a,b for EDM). The clone model trained on this data fails to capture any meaningful decision boundary (Fig. 1c for EDM) and produces poor accuracy on the classification task, making MS attacks harder to carry out. In summary, our paper makes the following key contributions:

1. We propose a novel *Diversity Loss* function to train an *Ensemble of Diverse Models* (EDM) that produce dissimilar predictions on OOD data.

2. We propose using EDM to defend against model stealing attacks. Our defense creates discontinuous predictions for the adversary's OOD queries, making MS attacks harder to carry out, without causing degradation to the model's accuracy on benign queries.

3. We develop a DNN-based perceptual hash function, which produces the same hash value even with large changes to the input, making adaptive attacks harder to carry out.

## 2    RELATED WORK

**Model Stealing Attacks:** MS Attacks fall into one of three categories: (i) parameter stealing, (ii) hyper-parameter stealing and (iii) functionality stealing attacks. Parameter stealing (Tramèr et al., 2016; Lowd & Meek, 2005) and hyperparameter stealing attacks (Wang & Gong, 2018; Oh et al., 2019) focus on inferring the exact model parameters or hyperparameters used in the architecture/training of the target model respectively. Our work focuses on defending against functionality stealing attacks, where the adversary's goal is to train a clone model that achieves high accuracy on a given task. Since the training data of the target is usually unavailable, attacks leverage alternate forms of data to query the target. Orekondy et al. (2019a); Correia-Silva et al. (2018) use datasets from related problem domains, while other attacks (Papernot et al., 2017; Juuti et al., 2019; Roberts et al., 2019; Kariyappa et al., 2020) have proposed crafting synthetic examples to query the target to perform model stealing.

**Model Stealing Defenses:** Existing defenses that have been proposed to defend against model functionality stealing involve perturbing the output predictions to prevent the adversary from having reliable access to the predictions of the target model. Perturbation-based defenses modify the output of the model $y = f(x)$ to produce a perturbed output $y' = y + \delta$. These defenses can be further sub-categorized into *accuracy-preserving* and *accuracy-constrained* defenses.

Accuracy-Preserving Defenses (Lee et al., 2018) ensure that the class-labels of the perturbed predictions $y'$ are the same as the original prediction i.e. $argmax(y') = argmax(y)$. Since the class-labels are unchanged, the benign accuracy of the model is preserved. In contrast, accuracy-constrained defenses (Kariyappa & Qureshi, 2020; Orekondy et al., 2019b) allow for the usage of a higher level of perturbation, trading off benign accuracy for increased security against MS attacks.

This work presents the first defense that is not reliant on adding perturbations to the predictions of the model. Instead, we aim to create an entirely new class of ensemble-based models that is inherently hard to steal with a data-limited attack, due to the discontinuous nature of the predictions produced by the model. Our method has minimal impact on the benign accuracy and offers a much higher security against MS attacks compared to a single model. Additionally, our model can be used in conjunction with perturbation-based defenses to further improve robustness against MS attacks.

## 3    PRELIMINARIES

### 3.1    ATTACK: OBJECTIVE AND CONSTRAINTS

The goal of the attacker is to maximize the accuracy of a clone model $C$ on a test set $\mathcal{D}_{test}$ i.e. $\max_{\theta_C} \mathbb{E}_{x,y \sim \mathcal{D}_{test}}[Acc(C(x;\theta_C),y)]$. MS attacks use the principle of knowledge distillation to train a clone model. First, the attacker queries the target model using inputs $\{x_i\}_{i=1}^{n}$ to obtain the target's predictions $\{y_i = T(x_i)\}$. The labeled dataset $\{x_i, y_i\}$ created this way can then be used to train a clone model $C$, completing the attack. The predictions returned from $T$ can either be *soft-label prediction*, where $T$ returns a probability distribution over the output classes, or *hard-label predictions*, where only the argmax class-label is returned. We consider attacks under both prediction types in this paper. Since there is a fixed cost associated with querying $T$, we assume that the number of queries is upper bounded by a query budget $Q$.

**Data Constraints**: In most real-world settings, the adversary does not have access to the dataset used to train the target model. A data-limited attacker can use alternate forms of data to carry out model stealing attacks. KnockoffNets (Orekondy et al., 2019a) uses data from an alternate surrogate dataset, which has semantic similarities with the target model and Jacobian-Based Dataset Augmentation (Papernot et al., 2017) uses synthetic datasets to query the target model.

### 3.2    DEFENSE: OBJECTIVE AND CONSTRAINTS

The goal of the defense is to minimize the accuracy of the clone model i.e. $\min \mathbb{E}_{x,y \sim \mathcal{D}_{test}}[Acc(C(x;\theta_C),y)]$. The defense is allowed to make changes to the target model or the model predictions. However, we require the defense to have minimal impact on the accuracy of the model for benign inputs to preserve the utility of the model. The defender is unaware of the kind of attack being used, or the dataset used by the adversary to query the target

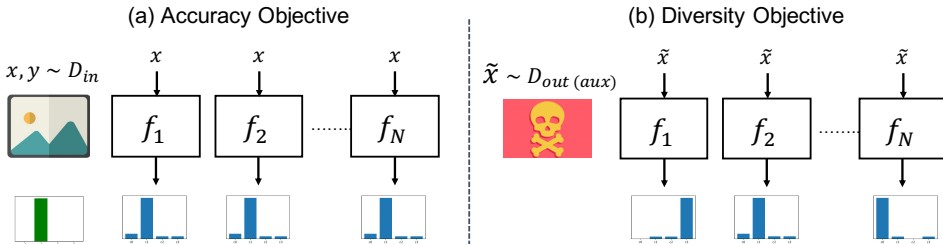

Figure 2: Models in EDM are jointly trained on two objectives: (a) *Accuracy Objective* ensures that the models make correct predictions for in-distribution data. (b) *Diversity Objective* encourages the models to make diverse decisions for auxiliary out-of-distribution data.

model. In addition to existing attacks, the defense also needs to be effective against adaptive attacks that are tailored for the defense.

# 4 OUR PROPOSAL: ENSEMBLE OF DIVERSE MODELS

This paper proposes a novel type of model ensemble called *Ensemble of Diverse Models* (EDM), which is significantly more robust to MS attacks compared to a single DNN model. EDM consists of a set of *diverse* models, which produce dissimilar predictions for OOD data. By using an input-based hash function, EDM selects a single model from the ensemble to service each query. Since the models are trained to produce dissimilar predictions, EDM outputs predictions that are highly discontinuous in the input space for OOD data. The complexity of the discontinuous predictions cause the clone model trained on these predictions to generalize poorly on in-distribution examples, making MS attacks less effective. We explain how EDM models can be trained and deployed for inference in this section.

## 4.1 TRAINING A DIVERSE ENSEMBLE

EDM leverages an ensemble of *diverse* models to produce discontinuous predictions. Let $\{f_i\}_{i=1}^{i=N}$ denote the ensemble of models used in EDM with model parameters $\{\theta_i\}_{i=1}^{i=N}$. These models are jointly trained with a two-fold training objective using an in-distribution dataset of labeled examples $\mathcal{D}_{in}$ and an out-of-distribution dataset of unlabeled examples $\mathcal{D}_{out}$ as explained below:

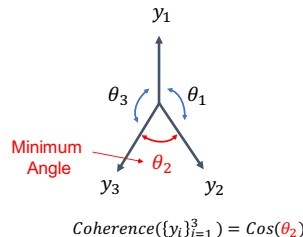

$$Coherence(\{y_i\}_{i=1}^3) = Cos(\theta_2)$$

*1. Accuracy objective:* Models in the ensemble should be trained to produce high accuracy for the examples in the training dataset of in-distribution examples $\mathcal{D}_{in}$ as shown in Fig. 2a. For a multi-class classification problem, let $\{\hat{y}_i = f_i(x; \theta_i)\}_{i=1}^N$ be the prediction probabilities of the models where $(x, y) \in \mathcal{D}_{in}$ is the input, label pair. We use the cross-entropy loss averaged across all models: $\frac{1}{N}\sum_{i=1}^N \mathcal{L}_{CE}(\hat{y}_i, y)$ to train the models in the ensemble to achieve high accuracy on the in-distribution training data.

Figure 3: Coherence measures the cosine of the smallest angle made by any pair of vectors in a set.

*2. Diversity objective:* The diversity objective requires the models in the ensemble to produce dissimilar predictions for examples in the out of distribution dataset $\mathcal{D}_{out}$ as depicted in Fig. 2b. If $\{\tilde{y}_i = f_i(\tilde{x}; \theta_i)\}_{i=1}^N$ are the output probabilities produced by the ensemble for an OOD input $\tilde{x} \in \mathcal{D}_{out}$, then the diversity objective requires the set of output vectors $\{\tilde{y}_i\}_{i=1}^N$ to be misaligned. We use the coherence metric (Tropp, 2006) to measure the alignment of the output probability vectors. Coherence of a set of vectors measures the maximum cosine similarity (CS) i.e. cosine of the smallest angle, between all pairs of probability vectors in the set as shown in Fig. 3. Coherence can be computed as follows:

$$coherence(\{\tilde{y}_i\}_{i=1}^N) = \max_{\substack{a,b\in\{1,..,N\}\\a\neq b}} CS(\tilde{y}_a, \tilde{y}_b). \tag{1}$$

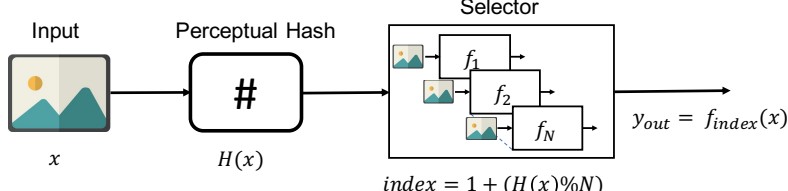

$$index = 1 + (H(x)\%N)$$

Figure 4: EDM uses an input-based hash $H(x)$ to select the model that is used to service an input query $x$. A perceptual hashing algorithm is used to prevent adaptive attacks.

In order to increase the misalignment of probability vectors, we need to reduce the value of coherence. Unfortunately, the $max$ function in Eqn 1 makes coherence non-smooth, which precludes the use of first-order methods to optimize the value of coherence. To overcome this limitation, we use the LogSumExp function to get a smooth approximation of the $max$ function that allows it to be used in first-order optimization algorithms. We term the resulting loss function as the *diversity loss*:

$$DivLoss(\{\tilde{\boldsymbol{y}}_i\}_{i=1}^N) = \log \left( \sum_{1 \le a < b \le N} \exp(CS(\tilde{\boldsymbol{y}}_a, \tilde{\boldsymbol{y}}_b)) \right). \tag{2}$$

Models in the ensemble are jointly trained on the combination of accuracy and diversity objectives:

$$\mathcal{L} = \mathbb{E}_{x,y \sim \mathcal{D}_{in}, \tilde{x} \sim \mathcal{D}_{out}} \left[ \left( \frac{1}{N} \sum_{i=1}^N \mathcal{L}_{CE}(\hat{\boldsymbol{y}}_i, \boldsymbol{y}) \right) + \lambda_D \cdot DivLoss(\{\tilde{\boldsymbol{y}}_i\}_{i=1}^N) \right], \tag{3}$$

$$where \ \hat{y}_i = f_i(x), \ \tilde{y}_i = f_i(\tilde{x}).$$

Here $\lambda_D$ is a hyperparameter that dictates the ratio of importance between the diversity and accuracy objectives. Note that the loss function described above requires access a dataset of labeled in-distribution examples $\mathcal{D}_{in}$ as well as a dataset of out-of-distribution examples $\mathcal{D}_{out}$. Since we do not know the OOD samples that is used by the adversary a-priori, we make use of an auxiliary OOD dataset to train EDM. Similar use of auxiliary OOD datasets have been explored in anomaly detection literature (Hendrycks et al., 2018) to detect unseen anomalies. Our experiments in Section 5.3 suggest that the diversity objective generalizes to the unseen OOD dataset used by the adversary.

## 4.2 EDM INFERENCE

The objective of EDM is to produce predictions that are discontinuous in the input space. EDM produces discontinuous predictions by selecting a model in the ensemble based on the input to service each request as shown in Fig. 4. Given an input query $x$, EDM uses a hashing function $H$ to compute a hash of the input, from which the index of the model that is used to service the query can be computed: $index = 1 + (H(x)\%N)$. The output prediction is given by $y_{out} = f_{index}(x)$.

### 4.2.1 CHOOSING THE HASH FUNCTION

To ensure that an adversary is unable to launch an adaptive attack to game the hash function, we select a hash function that satisfies the following properties:

1. *Security:* The hash function needs to be kept secret from the adversary to avoid the adversary from computing the hash and figuring out which model was used to service a given input.

2. *Transformation invariance:* While the hash function should map different inputs to different hash values, we want the hash value to be invariant to small transformations in the input images such as scaling, rotation, translation, warping, contrast adjustment, etc. This invariance property is necessary to prevent an adversary from using slightly modified versions of the original image to change the hash value and query a different model. Without the invariance property, the adversary can query all the models in the ensemble and get the average prediction for a given input, which is continuous in nature and allows for more effective model stealing.

Cryptographic hash function would be unsuitable for our application as they lack transformation invariance. Perceptual hashing (pHash) (Zauner, 2010) are a class of hash functions that ensure that

perceptually similar inputs map to the same/similar hash values by using transformation invariant features (like discrete cosine transform), making them better suited for our application. Empirically, we found existing pHash schemes to be less robust to various transformations of the inputs (discussed in Appendix A.2). To solve this problem, we propose a novel DNN-based perceptual hashing function that outputs the same hash value even with large changes to the input. Our insight is that for any DNN that is trained on a classification problem involving natural images, the output prediction of the DNN is typically invariant to transformations of the input that retain perceptual similarity. Given a DNN $f_H$ that is trained on a secret $M$-class classification problem, we can use the $argmax$ predictions to compute a hash of the input: $H(x) = argmax(f_H(x))$. To ensure that the hash function uniformly maps inputs to different indices, we increase the output space of the hash function by subdividing each class into $m$ sub-classes (achieving a total of $M * m$ possible outputs) using the following function: $H(x) = m \cdot argmax(f_H(x)) + \left\lfloor \frac{M \cdot max(H(x)) - 1}{M - 1} \cdot m \right\rfloor$. We use a small value for $m$ ($m = 2$) in our experiments to ensure transformation invariance is preserved. Additionally, a relatively small model can be used for $f_H$ to reduce computational overheads. The model architecture and the dataset used to train $H$ are unknown to the adversary, making the output of the hash function hard to estimate. The security of the hash function can further be improved by training it on a custom dataset that is not available in the public domain.

## 5 EXPERIMENTAL RESULTS

### 5.1 SETUP

**Victim models and datasets:** We evaluate our defense by comparing the clone accuracy achieved by various attacks on EDM and an undefended baseline model. A lower clone accuracy implies better security. We use DNNs trained on various image-classification datasets listed in Table 3 as the target models for performing MS attacks. For each dataset, we train two target models 1. *undefended baseline:* A single model trained on $\mathcal{D}_{in}$ using cross entropy loss. 2. *EDM*: An ensemble of models

Table 1: Datasets, model architecture and accuracies of the target models.

| Dataset | Model Arch. | Accuracy (%) | |
|---|---|---|---|
| | | Undef. | EDM |
| MNIST | Conv3 | 99.49 | 99.5 |
| FashionMNIST | Conv3 | 91.02 | 91.01 |
| CIFAR-10 | Wres16-4 | 94.83 | 94.35 |
| CIFAR-100 | Wres16-4 | 74.87 | 74.83 |
| Flowers-17 | ResNet-18 | 95.15 | 95.22 |

trained on the EDM loss function (Eqn. 3). The dataset, model architecture and test accuracies obtained by the undefended target and EDM targets are shown in Table 1. Note that, despite the additional *diversity objective*, the accuracy of EDM is comparable to the undefended baseline model.

**Attacks:** We consider an adversary who is interested in stealing model functionality by training a high-accuracy clone-model on the predictions of the target model. We assume that the adversary does not have access to the entirety of the training dataset that was used to train the target model. The number of queries that the adversary can make to the target model is limited by the query budget $Q$ (We use $Q = 50000$ for our evaluations). Additionally, we assume that the adversary is aware of the target model's architecture and use a clone model with the same architecture as the target. We consider three recent MS attacks to evaluate our defense:

1. *KnockoffNets* (Orekondy et al., 2019a) performs model stealing by using a surrogate dataset to query the target model and training a clone model $C$ on the predictions of the target model. We use FashionMNIST, MNIST, CIFAR100, CIFAR10, and Indoor67 (Quattoni & Torralba, 2009) as the surrogate datasets for MNIST, FashionMNIST, CIFAR-10, CIFAR-100, and Flowers-17 (Nilsback & Zisserman, 2006) models respectively. The clone model is trained using an SGD optimizer with an initial learning rate of 0.1 for 50 epochs with cosine-annealing learning-rate schedule.[1]

2. *Jacobian Based Dataset Augmentation (JBDA)* (Papernot et al., 2017) uses synthetic data to query the target model. JBDA starts with a small set of in-distribution "seed" examples $\{x_i\}$. The attack iteratively trains the clone model by performing the following steps: (i) Obtain a labeled dataset $\{x_i, y_i\}$ by querying the target model (ii) Train the clone model on the labeled dataset (iii) Augment the dataset with synthetic examples $x_i'$ by perturbing the original input $x_i$ to change the prediction of the clone model by using the Jacobian of the loss function: $x_i' = x_i + \beta sign\left(\nabla_x \mathcal{L}\left(C\left(x_i; \theta_c\right), y_i\right)\right)$.

---

[1]For Flowers-17, we use a clone model that is pre-trained on Imagenet with an initial learning rate of 0.001.

Table 3: Comparing the clone accuracy produced by attacking an undefended baseline and EDM target model for various model stealing attacks under soft and hard label settings. Attacks targeting EDM produce significantly lower clone accuracies compared to an undefended model in most cases.

| | Dataset | KnockoffNets | | JBDA | | JBDA-TR | |
|---|---|---|---|---|---|---|---|
| | | Undef. (%) | EDM (%) | Undef. (%) | EDM (%) | Undef. (%) | EDM (%) |
| **Soft-Label** | MNIST | 90.18 (0.91x) | **51.34 (0.52x)** | 88.48(0.89x) | **79.04(0.79x)** | 89.82(0.90x) | **73.08(0.73x)** |
| | FashionMNIST | 34.76 (0.38x) | **15.28 (0.17x)** | 73.28(0.80x) | **72.34(0.79x)** | **73.27(0.80x)** | 74.17(0.81x) |
| | CIFAR-10 | 85.39 (0.90x) | **68.50 (0.73x)** | 22.03(0.23x) | **22.01(0.23x)** | 21.44(0.23x) | **20.96(0.22x)** |
| | CIFAR-100 | 53.04 (0.71x) | **41.16 (0.55x)** | 4.09 (0.05x) | **3.69 (0.05x)** | 4.45 (0.06x) | **4.22 (0.06x)** |
| | Flowers-17 | 69.85 (0.72x) | **30.15 (0.31x)** | 11.76(0.12x) | **11.40(0.12x)** | 20.59(0.21x) | **18.75(0.19x)** |
| **Hard-Label** | MNIST | 61.20 (0.62x) | **41.25 (0.41x)** | 81.53(0.82x) | **72.64(0.73x)** | 80.62(0.81x) | **73.18(0.74x)** |
| | FashionMNIST | 28.60 (0.31x) | **15.64 (0.17x)** | 70.97(0.78x) | **66.50(0.73x)** | 70.23(0.77x) | **69.46(0.76x)** |
| | CIFAR-10 | 73.64 (0.78x) | **54.98 (0.58x)** | 18.99 (0.20x) | 20.08 (0.21x) | 24.47 (0.26x) | **18.78 (0.20x)** |
| | CIFAR-100 | 28.50 (0.38x) | **24.69 (0.33x)** | 4.23(0.06x) | **3.75(0.05x)** | 5.24(0.07x) | **3.38(0.05x)** |
| | Flowers-17 | 56.25 (0.58x) | **28.68 (0.30x)** | 11.76(0.12x) | **11.03(0.11x)** | **15.07(0.16x)** | 16.54(0.17x) |

We use 6 rounds of data augmentation with the value of $\beta$ set to 0.2 for the JBDA attack. The clone model is trained for 10 epochs in each round with the Adam optimizer with a learning rate of 0.001.

3. *JBDA-TR* is a variation of JBDA proposed by Juuti et al. (2019) that produces targeted misclassification from the clone model by using the gradient of the loss function to move towards a random target class $(y_T)$: $x'_i = x_i - \lambda sign \left( \nabla_x \mathcal{L} \left( C \left( x_i; \theta_c \right), y_T \right) \right)$

**EDM Defense:** For the EDM defense, we use an ensemble of 5 models that are jointly trained on the accuracy and diversity objectives by using the loss function in Eqn. 3. The value of $\lambda_D$ was selected with the constraint of having the degradation in benign accuracy of the target model be less than 0.5%. EDM also requires an auxiliary OOD dataset for the diversity objective. We use KMNIST (Clanuwat et al., 2018) for MNIST and FashionMNIST, Tiny-Images (Torralba et al., 2008) for CIFAR10 and CIFAR100, and ImageNet1k for Flowers-17 as the auxiliary OOD datasets. For the DNN-based perceptual hashing we use $M = 10$ and $m = 2$. KMNIST is used as the secret classification task for MNIST and FashionMNIST. SVHN is used for the rest of the datasets. We use a simple 3-layer convolutional neural network trained on the secret classification task as the hash function $f_H$ to keep the computational overheads low.

## 5.2 RESULTS

Our results comparing the efficacy of various MS attacks targeting an undefended baseline and EDM targets for models trained on different classification tasks are shown in Table 3. In each case, we report the accuracy of the clone model along with the normalized clone accuracy computed with respect to the accuracy of the target model. Our results show that the clone accuracies produced by the KnockoffNets attack are much lower for EDM compared to the baseline model. With the JBDA and JBDA-TR attacks, the benefits of EDM are less pronounced as these attacks use perturbed versions of in-distribution examples to query the target, which may not produce large discontinuities in the predictions. However, JBDA and JBDA-TR only work well on simpler datasets (MNIST, FashionMNIST) and do not produce good accuracies on higher dimensional datasets. Overall, we find that the discontinuous predictions produced by EDM can reduce the effectiveness of model stealing attacks for most attacks across multiple target models, offering up to a 39.7% reduction in clone accuracy compared to an undefended baseline model for existing attacks. Furthermore, our defense is also effective against adaptive attacks that are tailored for our defense (see Appendix A.1).

**Comparison with prior work:** We compare the clone accuracy obtained from our method with the adaptive misinformation (AM) defense (Kariyappa & Qureshi, 2020). AM aims to output incorrect predictions for the adversary's queries by adaptively perturbing the predictions of the model for OOD inputs. Unlike our work, AM trades off the benign accuracy for improved security. We compare the clone accuracy obtained from KnockoffNets attack between EDM and AM, with equivalent benign accuracy in Table 2. Our results show that EDM is able to outperform AM in most datasets with up to 32.16% lower clone accuracy compared to the AM defense. We see comparable performance with AM in case of Flowers-17 dataset.

Table 2: Comparison of EDM with Adaptive Misinformation defense

| Dataset | Clone Accuracy (%) | |
|---|---|---|
| | AM | EDM |
| MNIST | 83.5 (0.84x) | **51.34 (0.52x)** |
| FashionMNIST | 27.5 (0.30x) | **15.28 (0.17x)** |
| CIFAR-10 | 77.2 (0.82x) | **68.50 (0.73x)** |
| CIFAR-100 | 51 (0.67x) | **41.16 (0.55x)** |
| Flowers-17 | **27.2 (0.29x)** | 30.15 (0.31x) |

### 5.3 ANALYSIS

**Impact of Diversity Loss on Coherence:** To quantify the efficacy of the *diversity loss* in producing dissimilar predictions, we compare the CDF of coherence values obtained for the predictions of an EDM model trained with diversity loss and a regular ensemble (with the same number of models) trained without diversity loss for the adversary's queries used in the KnockoffNets attack. From our results in Fig. 5, we find that the models in EDM produce much lower coherence values compared to a regular ensemble. Additionally, these plots show that the models in EDM, which are trained to produce

Table 4: Comparing the clone accuracy produced by KnockoffNets by attacking a regular ensemble and a diverse ensemble.

| Dataset | Clone Accuracy (%) | |
|---|---|---|
| | Reg. Ens. | Div. Ens. |
| MNIST | 88.04 (0.88x) | **51.34 (0.52x)** |
| FashionMNIST | 30.11 (0.33x) | **15.28 (0.17x)** |
| CIFAR-10 | 84.25 (0.89x) | **68.50 (0.73x)** |
| CIFAR-100 | 50.44 (0.67x) | **41.16 (0.55x)** |
| Flowers-17 | 54.78 (0.57x) | **30.15 (0.31x)** |

low coherence values on an auxiliary OOD dataset, can generalize to unseen OOD data used by the adversary to carry out model stealing.

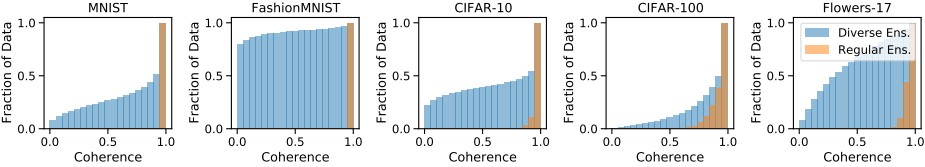

Figure 5: Comparing the CDF of coherence values between a regular ensemble (trained without diversity loss) and a Diverse Ensemble (trained with diversity loss). Diverse ensemble produces much lower coherence values on the adversary's OOD data, indicating dissimilar predictions.

**Impact of Diversity Loss on Clone Accuracy:** Models trained without the diversity objective can also be used in our proposal to produce discontinuous predictions. However, such models produce correlated predictions, making them easier to steal. We show the importance of using the diversity objective by comparing the clone accuracies obtained by attacking two ensembles– 1. Regular Ensemble (trained without diversity loss) 2. EDM (trained with diversity loss)– using the KnockoffNets attack. Our results in Table 4 show that regular ensembles produce higher clone accuracies compared to the diverse ensemble, emphasising the importance of training with diversity loss.

## 6 DISCUSSION ON ADAPTIVE ATTACKS

An adversary who is aware of our defense can potentially modify the attack to improve the efficacy of MS. Fundamentally, an adversary can overcome our defense if the discontinuous predictions of EDM can be converted to a continuous function. We discuss two adaptive attack strategies and explain why such adaptive attacks are hard to carry out.

1. Attacking a single model: Since the predictions of any individual model in EDM are continuous, the attacker can try to obtain the predictions of a single model and use these predictions to train the clone model. However, since the hash function is kept secret, it is not possible for the adversary to know which model was used to service a given request, making such attacks difficult to execute.

2. Attacking the Ensemble: A second strategy is to somehow obtain the average predictions of all the models in the ensemble for a given input. Since the average prediction is continuous, this can potentially be used to train a high-accuracy clone model. To obtain the average predictions, the adversary needs to make small perturbations to the input that cause the input to get hashed to a different model. However, the use of perceptual hashing in our defense makes such perturbation-based averaging harder to perform. Perceptual hashing ensures that small perturbations to the input result in the input mapping to the same hash. For large perturbations that do map to a different hash, the input no longer resembles the original input, making the averaging process meaningless. Furthermore, we note that even if the hash function was known to the adversary, such an attack would be costly, requiring an $\mathcal{O}(N)$ increase in the query budget to carry out. Additionally, querying the target model with a set of perturbed inputs opens up the attack to detection using methods such as PRADA (Juuti et al., 2019), which can limit the applicability of such adaptive attacks. We refer the reader to Appendix A.1 for a detailed analysis of this adaptive attack strategy.

## 7 CONCLUSION

This paper proposes *Ensemble of Diverse Models*, a novel type of model ensemble that is inherently robust to MS attacks due to the discontinuous nature of the model's predictions. The models in EDM are trained to produce dissimilar predictions on OOD data by using the *diversity loss* function. Our defense selects a different model from the ensemble to service different inputs using an input-based hash function, producing predictions that are highly discontinuous in the input space for the adversary's OOD inputs. Attacks that use these discontinuous predictions to train a clone model end up with a bad quality clone model, reducing the efficacy of model stealing attacks. Through our empirical evaluations, we show that EDM can reduce the clone accuracy by up to 39.7%, with minimal degradation ($< 0.5\%$) in the accuracy of the target model on benign inputs. Additionally, EDM has low computational overheads, making it a practical solution to defend against MS attacks.

## 8 ACKNOWLEDGEMENTS

We thank Ryan Feng, Neal Mangaokar and Ousmane Dia for their valuable feedback. This work was partially supported by a gift from Facebook and is partially based upon work supported by the National Science Foundation under grant numbers 1646392 and 2939445 and under DARPA award 885000. We also thank NVIDIA for the donation of the Titan V GPU that was used for this research.

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

# A APPENDIX

## A.1 EVALUATING ADAPTIVE ATTACKS AGAINST EDM

As discussed in section 6, one possible strategy to attack EDM is to obtain the average predictions of all the models in the ensemble and train the clone model on this average prediction. To explain this adaptive attack, consider an input $x$ for which we would like to compute the average prediction of all the models in the ensemble: $\bar{y} = \frac{1}{N} \sum_{i=1}^{N} f_i(x)$, where $\{f_i\}$ are the set of all the models in EDM. EDM uses an input-based hash: $index(x) = 1 + H(x)\%N$ to select the model $f_{index}$ that is used to service $x$. If the hash function $H$ was known, the adversary could modify the original input $x$ to obtain a perturbed input $x' = x + \delta$ that maps to a different hash value such that $index(x') \neq index(x)$. By querying EDM with multiple perturbed inputs $\{x'_j\}$ that produce different values for $index(x_j)$, the adversary can obtain an approximation of the average predictions $\bar{y} = \frac{1}{K} \sum_{j=1}^{K} f_{EDM}(x'_j)$. This approximation estimates the true average prediction well as long as the size of the perturbation is small enough to not change the predictions of the model significantly i.e. $f_i(x'_j) \approx f_i(x)$. Once the average predictions are obtained, a labeled dataset $\{x, \bar{y}\}$ can be constructed using these average predictions to train the clone model.

However, the hash function is unknown to the adversary, which gives rise to two challenges. First, evaluating the value of the $index(x)$ is not possible and second, finding small perturbations that would change the value of $index(x'_j)$ is hard. One approach to solving this problem is by averaging the predictions of the EDM over multiple inputs with large random perturbations $\{x'_j\}_{j=1}^{k}$ such that $\bar{y} = \frac{1}{K} \sum_{j=1}^{K} f_{EDM}(x'_j)$.

Table 5: Evaluation of adaptive attack against EDM using multiple perturbed inputs

| Dataset | Undef. | EDM | EDM (adaptive) |
|---|---|---|---|
| MNIST | 90.18 (0.91x) | 51.34 (0.52x) | 72.00 (0.72x) |
| FashionMNIST | 34.76 (0.38x) | 15.28 (0.17x) | 15.02 (0.16x) |
| CIFAR-10 | 85.39 (0.90x) | 68.50 (0.73x) | 63.89 (0.68x) |
| CIFAR-100 | 53.04 (0.71x) | 41.16 (0.55x) | 50.98 (0.68x) |
| Flowers-17 | 69.85 (0.72x) | 30.15 (0.31x) | 54.78 (0.57x) |

Large perturbations increases the likelihood that inputs map to different indices. Averaging over a a large set of such perturbed inputs improves the estimation of the average prediction. We evaluate the efficacy of such an adaptive attack empirically. We perform an adaptive version of KnockoffNets attack by using the average predictions over $K = 5$ perturbed inputs to train the clone model. Note that such an averaging attack requires a 5-fold increase in the query budget to carry out. We use a combination of random affine transforms (rotation, translation, scaling, shear) and random changes to brightness, contrast, color and hue of an image to obtain the perturbed inputs $\{x'_j\}_{j=1}^{k}$ from the original input $x$. Our results comparing the clone accuracy obtained from this adaptive attack compared to the non-adaptive attack on EDM and the single undefended model are shown in Table 5. Our results show that, while the clone accuracy improves with the adaptive attack compared to the non-adaptive attack on EDM, it still falls short of the clone accuracy obtained by attacking a single model. The improvement in clone accuracy can be attributed to the averaging of the decisions from multiple models. There are two reasons why the adaptive attack still falls short of the accuracy with a single model: 1. The predictions of the model could be different for the perturbed input i.e $f_i(x'_j) \not\approx f_i(x)$, which gives rise to an approximation error. 2. The DNN-based perceptual hashing algorithm ensures that most of the perturbed inputs map to the original index, preventing proper averaging across the predictions of all models.

## A.2 EFFICACY OF DNN-BASED PHASH

An ideal perceptual hash function would generate the same hash value for the original $(x)$ and perturbed inputs $(x')$. To understand the efficacy of our proposed pHash algorithm, we plot the sorted histogram of index values: $index(x) = 1 + H(x)\%N$, generated from our hashing function for 10000 randomly perturbed images. We aggregate the histogram of sorted indices for 1000 random examples in the training set of the CIFAR-10 dataset. We compare the performance of our proposed hash function against a Discrete Cosine Transform (DCT) based perceptual hash Zauner (2010) and an SHA-1 cryptographic hash. From our results in Fig. 6, we can see that our DNN-based pHash produces a histogram that more skewed compared to the other hashing schemes. This implies that our hashing function has better transformation invariance property as it maps most of the perturbed inputs to the same index, making it suitable for perceptual hashing.

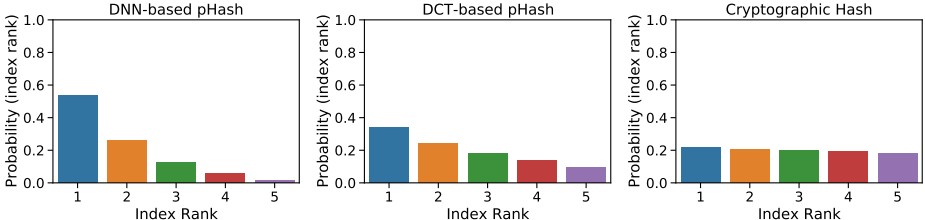

Figure 6: Sorted histogram of index values outputs by the DNN-based pHash for perturbed inputs in CIFAR-10

Table 6: Evaluation of EDM against KnockoffNets with a larger clone model. Using a larger clone model yields similar clone accuracies as the original clone model.

| Dataset | Undef. (%) | EDM-Orig. | EDM-Large | Orig. Model | Large Model |
|---|---|---|---|---|---|
| MNIST | 90.18 (0.91x) | 51.34 (0.52x) | 53.10 (0.53x) | Conv3 | ResNet-20 |
| FashionMNIST | 34.76 (0.38x) | 15.28 (0.17x) | 15.73 (0.17x) | Conv3 | ResNet-20 |
| CIFAR-10 | 85.39 (0.90x) | 68.50 (0.73x) | 67.52 (0.72x) | Wres16-4 | Wres22-4 |
| CIFAR-100 | 53.04 (0.71x) | 41.16 (0.55x) | 43.45 (0.58x) | Wres16-4 | Wres22-4 |
| Flowers-17 | 69.85 (0.72x) | 30.15 (0.31x) | 9.93 (0.11x) | ResNet-18 | ResNet-32 |

## A.3 COMPATIBILITY OF EDM WITH OTHER DEFENSES

Our paper proposes to use the discontinuous predictions produced by an ensemble of diverse models to defend against MS attacks. Prior works on MS defense have mainly focused on perturbing the output predictions to prevent the adversary from reliably accessing the predictions of the model. Since EDM does not modify the output predictions of the model, our defense can be combined with existing perturbation based defenses (Orekondy et al., 2019b; Lee et al., 2018; Kariyappa & Qureshi, 2020) to further improve the security of the model against MS attacks.

## A.4 COMPUTATIONAL COST

Compared to an undefended baseline model, EDM requires training multiple models, increasing the training cost by a factor of $N$. However, during inference, only a single model from the ensemble is selected and evaluated, making the inference cost comparable with the baseline. The additional compute associated with evaluating the hash function is negligible compared to the cost of evaluating the DNN used for the main classification task. For instance, the Conv3 model used in the hash function for CIFAR-10 only requires a $\sim 2.5\%$ increase in the number of flops compared to evaluating a single Wres16-4 network. EDM also requires a factor of $N$ increase in memory costs to store the ensemble. However, in a datacenter setting with multiple machines running the same inference task, the memory cost can be avoided by spreading the models of the ensemble across multiple machines, with each input being routed to the selected model for inference. The negligible impact on benign accuracy and minimal computational overheads make EDM a practical solution to defend against MS attacks.

## A.5 KNOCKOFFNETS WITH A LARGER CLONE MODEL

In the main paper, we evaluated our defense with attacks that used clone models with the same architecture as the target model. In this section, we report results for the KnockoffNets attack by using a larger model as the clone. We compare the clone accuracy obtained by using the original and the larger clone models in Table 6. Our results show that clone accuracy obtained with a larger model architecture is similar to the clone accuracy obtained with the original model architecture.

## A.6 EVALUATIONS OF ATTACKS WITH A LARGER QUERY BUDGET

We used an attack query budget of 50K to evaluate our defense in the main paper. In this section, we report the results with an increased query budget of 100K queries ($2\times$). Note that, for the

Figure 7: Sorted histogram of index values outputs by the DNN-based pHash for perturbed inputs in CIFAR-10

KnockoffNets attack, in addition to the query budget, the attacker is bound by the availability of surrogate data used for the attack. Thus, for our evaluations, we stop querying the target model when we either run out of query budget or the surrogate data samples available to query the target model. For the JBDA and JBDA-TR attacks, to utilize the larger query budget, we increase the number of augmentation rounds from 6 to 10. Our results are reported in Table 7. We observe only a slight improvement in clone accuracy for KnockoffNets as the attack is limited by the availability of surrogate data (70K was the size of the largest surrogate dataset used). Even in the case of JBDA and JBDA-TR attacks, the clone accuracy only improves marginally as the improvement in clone accuracy plateaus after 6 augmentation rounds.[2] For all the attacks, EDM continues to have a significantly lower clone accuracy compared to the baseline, even with a larger query budget.

Table 7: Evaluations with a higher attack query budget of 100K queries. EDM continues to produce lower clone accuracy compared to the undefended baseline even with a higher query budget.

| | Dataset | KnockoffNets | | JBDA | | JBDA-TR | |
|---|---|---|---|---|---|---|---|
| | | Undef. (%) | EDM (%) | Undef. (%) | EDM (%) | Undef. (%) | EDM (%) |
| **Soft-Label** | MNIST | 92.15 (0.93x) | **51.5 (0.52x)** | 90.03(0.90x) | **72.46(0.73x)** | 91.38(0.92x) | **76.00(0.76x)** |
| | FashionMNIST | 35.66 (0.39x) | **18.68 (0.21x)** | 74.82(0.82x) | **74.66(0.82x)** | 76.35(0.84x) | **74.88(0.82x)** |
| | CIFAR-10 | 87.84 (0.93x) | **67.59 (0.72x)** | **22.90(0.24x)** | 23.09(0.24x) | 25.91(0.27x) | **21.02(0.22x)** |
| | CIFAR-100 | 53.73 (0.72x) | **44.60 (0.60x)** | **3.86(0.05x)** | 4.36(0.06x) | 4.99(0.07x) | **4.21(0.06x)** |
| | Flowers-17 | 69.85 (0.72x) | **30.15 (0.31x)** | 13.76(0.15x) | **12.87(0.14x)** | 20.66(0.21) | **17.28(0.19x)** |

## A.7 EVALUATIONS WITH ADAPTIVE MISINFORMATION DEFENSE

We compare the trade-off between defender accuracy (for benign examples) and clone accuracy of the Adaptive-Misinformation (AM) defense (Kariyappa & Qureshi, 2020) with our proposal. AM uses an out of distribution (OOD) detector to identify the adversary's OOD data and services such queries with incorrect predictions. By changing the value of the threshold used in the OOD detector, this defense offers a trade-off between benign accuracy and clone accuracy. In the case of EDM, one could expect a similar trade-off to exist by changing the value of the hyper-parameter $\lambda_D$ in Eqn.3. However, empirically we found that there is only a negligible loss in defender accuracy even with a large value of $\lambda_D$. This is because, the *Diversity Loss* term associated with $\lambda_D$ has a lower bound of 0. Empirically, we found that by increasing the value of $\lambda_D$, we can minimize the *Diversity Loss* term to 0 with a negligible loss in defender accuracy. Thus, instead of a trade-off curve, we denote EDM with a single trade-off point in our plot to compare against AM. Our results in Fig. 7 show that for the same defender accuracy, EDM yields a lower clone accuracy compared to AM.

---

[2]Our results are consistent with the observations made by Juuti et al. (2019).

