# OpenReview forum: "Protecting DNNs from Theft using an Ensemble of Diverse Models"
_ICLR.cc/2021/Conference — ICLR 2021 Poster_

### Official Review · AnonReviewer1 · 2020-10-27
**Blind Review**

**Rating:** 6
**Confidence:** 4

**Review:**

## Summary
- The paper proposes a defense against recent flavours of model stealing attacks by exploiting the insight that the recent effective attack query out of distribution examples to the victim model.
- The approach introduces discontinuities in the input-prediction space by (i) training an ensemble of models such that for a given OOD input, the predictions are randomized (ii) at inference time hashing an input to a particular input in the diverse ensemble.
- Evaluation on simple datasets (MNIST, Fashion, CIFAR) show that the approach is reasonably effective at defending and moreover without a significant degradation of the utility.

## Strengths

**1. Well-motivated problem**
- The defenses for model stealing are not as well-investigated as attacks. I appreciate the authors take a step towards addressing defenses; the initial results look promising.

**2. Insight**
- I like the insight used by the authors in the defense i.e., to generate discontinuities in the input-prediction space. This is well illustrated in Fig. 1.

**3. Writing**
- The paper is written well and is easy to follow.

## Concerns

### Major Concerns

**1. $D_{in}$ vs. $D_{out}$**
- It is a nice insight that adversarial users indeed rely on querying data from a different distribution. However, my first concern is that is also naturally true for benign users. After all, no one except the victim/defender has access to the training data distribution and all queried data (whether by adversary or benign users) is out of distribution.
- As a result, I wonder how applicable the defense is -- would the defense intentionally mispredict in non-IID query setups? For instance, when queried with a particular dog breed unseen during training? Or querying slightly translated digit images to standard MNIST classifier?
- More generally, the proposed approach requires drawing a strict line between in and out distribution which I find is not clearly analyzed (follow-up remarks in point 4).

**2. Diversity Objective**
- If I understand the training objective correctly (Fig. 2b, Eq. 3), each model $f_i$ in the ensemble is encouraged to generate a random prediction when queried with out-of-distribution (OOD) inputs. Moreover, in expectation, results in a uniform distribution over predictions on OOD data. For in-distribution data, the predictions ideally remain unchanged.
- As a result, my concern is that this appears to be simple OOD detection followed by misprediction/calibration in disguise. Consequently, I wonder if one can simply leverage existing advances in OOD detection to achieve the objective.
- Moreover the OOD implicitly performed here requires access to a corresponding dataset $D_{out}$ to model unknown OOD samples. This contrasts some existing OOD detection works (e.g., ODIN, Liang et al., ICLR '18) which shows success without explicitly modelling OOD data.

**3. Hashing**
- I am also concerned that the hashing strategy (Sec. 4.2) employed by the authors appears to be a weak link in the defense.
- Specifically, it appears that a robust defense requires the hashing function to consistently select the same model in the ensemble in spite of small changes in the input. However, going by Fig. 6 it does not necessarily seem to be the case.
- As a result, I wonder if an attacker can exploit the hashing function to recover whether the output is clean/poisoned -- such as by aggregating predictions over a set of transformed inputs.

**4. Evaluation**
- While the results look promising, I have two concerns here related to the evaluation.
- First, I would have preferred if the results in Table 3 were presented as curves (with defense vs. attacker accuracy). This would have provided some insights on how the defense performs w.r.t the important $\lambda$ hyperparameter in their objective. Additionally, it would also make the comparison with AM baseline fair, especially given that AM was evaluated on a curve and it is unclear which specific instance was used for comparison.
- Second, which also connecting to point 1, is that the paper's experimental setting assumes a significant discrepancy between the distributions e.g., Victim's $D_{in}$ = MNIST, $D_{out}$ = KMNIST, Attacker's data = FashionMNIST. Here, the data and semantic classes are completely disjoint. I wonder how the defense performs with other choices of attacker's data e.g., EMNIST which is a bit more similar to MNIST.


### Minor Concerns

**5. Experimental settings**
- I find missing some important experimental parameters that is not mentioned in the paper: how many images were queried by the attacker for results in Table 1? What was the value of $\lambda$ used?

**6. Subverting schemes**
- I was also disappointed that the authors do not demonstrate that the defense is robust to some simple schemes used by an attacker to bypass the proposed defense. For instance, connecting to point (3), by aggregating predictions over a set of transformed inputs.


### Nitpicks

**7. Some nitpicks**
- $\lambda$ is overloaded: in Eq. 3 and also for the JBDA attacks
- "AM trades off benign accuracy for improved security" -- would the proposed approach also trade this off by increasing the value of $\lambda$?

### Post-rebuttal updates
- Thanks for the detailed response and additional evaluation.
- Q1. Fair point -- it appears that defenses do assume attackers query from a reasonably different distribution. This defense does make this assumption explicit by including it in the training objective ($D_{out}$ in Eq. 3). But then again, it seems typical for OOD-based defenses (e.g., AM).
- Q3. This concern also generally connects to Q6. It would be nonetheless interesting to analyze scenarios where the attacker attempts to use some auxiliary knowledge to break the defense. This is also shared by some other reviewers.
- Q4. Thanks for presenting the curves. Assuming strictly non-overlapping OOD data does seem like a strong assumption though.
- Nitpick editorial comment: Please serif-based fonts for text in equations e.g. $argmax(\cdot) \rightarrow \text{argmax}(\cdot), index \rightarrow \text{index}$
- I am slightly increasing my rating.

---

> ### Author Response · Authors · 2020-11-18
> **Response**
>
> We thank the reviewers for their valuable and positive feedback. We are happy that the reviewers found our problem to be “well-motivated” and our idea to be “novel”. We are also glad that the reviewers found our paper to be “well-written” and our experimentation to be “extensive”. We hope to address the concerns raised by the reviewers below:
>
> **Q1. Din vs. Dout -- Would the defense intentionally mispredict in non-IID query setups?**
>
> EDM does not draw a strict line between in and out of distribution data (i.e. there is no step-function change in the misprediction/discontinuity of EDM beyond a threshold). Instead, the discontinuity/misprediction-rate produced by EDM increases gradually with the amount of “dataset shift” in the test data.
>
> The amount of dataset shift is assumed to be large in case of the adversary’s queries compared to the queries of a benign user (distribution of Translated-MNIST is much closer to MNIST than say FashionMNIST). Thus, we expect the misprediction rate to be high for the adversary’s queries compared to benign queries with a small dataset shift. This is a similar assumption that is made in other defenses (e.g., Prediction poisoning [1], Adaptive Misinformation [2]) as well.
>
> **Q2. Diversity Objective: My concern is that this appears to be simple OOD detection followed by misprediction/calibration in disguise. Consequently, I wonder if one can simply leverage existing advances in OOD detection to achieve the objective.**
>
> While our goal with EDM was to design a classifier that outputs discontinuous predictions on OOD data, our proposal also ends up generating random predictions on OOD data as a side effect. In fact, there is an existing proposal --Adaptive Misinformation (AM) [2] that is explicitly designed to output random predictions for OOD data, as suggested by the reviewer. AM uses an OOD detector to detect OOD queries and uses a misinformation model to output random predictions for such OOD queries. Our results comparing our defense with AM (Section 5.2 “comparison with prior work”) shows that our proposal offers better security compared to AM owing to the discontinuous predictions produced by our defense. We’ve also added more extensive evaluations comparing EDM with the trade-off curve of AM in Appendix A.7.
>
> **Q3. Hashing:  I wonder if an attacker can exploit the hashing function to recover whether the output is clean/poisoned -- such as by aggregating predictions over a set of transformed inputs.**
>
> Since the DNN used in the perceptual hashing function is not known to the adversary, the adversary needs a way to reliably ensure that the transformed input maps to a different model, which can be challenging.  Even if the adversary is able to figure out which inputs were poisoned, our defense would continue to outperform the baseline simply because a large fraction of the adversary’s data, which were deemed to have been poisoned, would have to be thrown out. Furthermore, the adversary would require a higher query budget to perform aggregation over multiple transformed inputs. Lastly, querying the target model with a set of transformed inputs opens up the attack to detection using methods such as PRADA (Juuti et al. 2019 [3]), which can limit the applicability of such adaptive attacks.
>
> **Q4. Evaluation: I would have preferred if the results in Table 3 were presented as curves (with defense vs. attacker accuracy)...Additionally, it would also make the comparison with AM baseline fair, .. I wonder how the defense performs with other choices of attacker's data e.g., EMNIST which is a bit more similar to MNIST**
>
> We’ve added more extensive evaluations comparing EDM with the trade-off curve of AM in Appendix A.7. The reason why we didn’t opt for defense vs attacker accuracy curves for EDM is that for most of the datasets, even with a very large value of λ, we saw negligible degradation in the defender accuracy of the defense. This is because our regularization term in Eqn 3 (DivLoss) gets minimized to its lowest value of 0 (0 is the lowest value of cosine similarity with positive valued vectors) with a large enough value of λ. Increasing λ further has no impact on clean accuracy.
>
> We did not evaluate with EMNIST as there is significant overlap between the classes of EMNIST and MNIST (0 and o, 1 and i/l, 9 and g/q, 8 and B, s and 5, z and 2). For the other datasets in our evaluations, we tried to pick surrogate datasets that are very similar to the target (CIFAR100 to attack CIFAR10, CIFAR10 to attack CIFAR100)
>
>
> [1] Tribhuvanesh Orekondy, Bernt Schiele, and Mario Fritz. Prediction poisoning: Utility-constrained defenses against model stealing attacks.
>
> [2] Sanjay Kariyappa and Moinuddin K Qureshi. Defending against model stealing attacks with adaptive misinformation.
>
> [3] Mika Juuti, Sebastian Szyller, Samuel Marchal, and N Asokan. Prada: protecting against dnn model stealing attacks.

---

### Official Review · AnonReviewer4 · 2020-10-27
**Good paper**

**Rating:** 7
**Confidence:** 3

**Review:**

Summary: The paper proposes a method to protect deep neural networks against model stealing. The propose defense trains an ensemble of classifiers using two losses one targeting accuracy and the other diversity of the ensemble. In particular, the trained classifiers are consistent on in-distribution data, but contradict each other on out-of-distribution data. The proposed defense does not affect the test accuracy of the victim model while it strongly limits the test accuracy of the clone model.

Despite the concerns below, I vote for accepting the paper. The addressed issue is relevant, the proposed method is interesting. The performance in the presented experiments is convincing and the writing is overall reasonably clear.

Concerns:
1. the method hinges on keeping the hash private ("security through obscurity")
2. it also hinges on the attacker not being able to query the API with in-distribution images
3. the authors argue that the inference runtime is not impacted by the method, however, the GPU memory requirements scale linearly with the size of the ensemble, which might be costly

Questions:
- it seems one could try to launch an adversarial attack on the hash, why should it fail?
- how far from the in-distribution images would one have to move to start hitting the effect of the diversity loss? As a user of an API, I have some knowledge about which images are in-distribution, why is using this information impractical?
- at the end of Sec 4.1, results on generalization to unseen out-of-distribution datasets are referred to, supposedly to be found in Sec 5.3, but I cannot find them there, where are they?
- the hash function should map to as many members of the ensemble as uniform as possible on both the in-distribution and out-of-distribution data, how is this achieved?

Typos:
- Table 3, the number for hard-label - Flowers-17 - JBDA-TR - EDM should most probably be 16.54 not 6.54, please check

---

> ### Author Response · Authors · 2020-11-18
> **Response**
>
>
> We thank the reviewers for their valuable and positive feedback. We are happy that the reviewers found our problem to be “well-motivated” and our idea to be “novel”. We are also glad that the reviewers found our paper to be “well-written” and our experimentation to be “extensive”. We hope to address the concerns raised by the reviewers below:
>
> **Q1. It seems one could try to launch an adversarial attack on the hash, why should it fail?**
>
> As the reviewer noted, the DNN used in the hash function is kept secret from the adversary. Furthermore, the adversary cannot access the output of the hash function which makes an adversarial attack on the hash function hard to carry out. One possible mode of attack is to make large perturbations to the input and hope that this changes the output of the hash function. We evaluate such an adaptive attack in Appendix A.1 and show that EDM continues to offer better security compared to our baseline. Additionally, querying the target model with a set of transformed inputs opens up the attack to detection using methods such as PRADA (Juuti et al. 2019 [1]), which can limit the applicability of such adaptive attacks.
>
>
>
> **Q2. How far from the in-distribution images would one have to move to start hitting the effect of the diversity loss? As a user of an API, I have some knowledge about which images are in-distribution, why is using this information impractical?**
>
> While there is no fixed boundary separating the in and out of distribution data, beyond which the effects of the diversity loss kick in, the amount of discontinuity offered by EDM increases as we move farther away from the in-distribution data. Similar to prior works (Adaptive Misinformation, Prediction Poisoning, Deceptive Perturbations ), our defense is intended for a data-limited adversary, who does not have access to a large collection of in-distribution data. The assumption here is that the number of in-distribution examples available to the adversary in and of itself is not sufficient to train a high-accuracy clone model. Hence the adversary either uses surrogate datasets (KnockoffNets attack) or perturbed versions of in-distribution data (JBDA, JBDA-TR) to carry out the attack.
>
> **Q3. At the end of Sec 4.1, results on generalization to unseen out-of-distribution datasets are referred to, supposedly to be found in Sec 5.3, but I cannot find them there, where are they?**
>
> Fig. 5 shows that a diverse ensemble produces lower coherence values on the adversary’s unseen OOD data compared to a regular ensemble. This shows that our diverse ensemble, which is trained to produce low coherence value on an auxiliary OOD dataset, can generalize to unseen OOD data used by the adversary. We have updated Section 5.3 to state this more explicitly.
>
> **Q4. The hash function should map to as many members of the ensemble as uniform as possible on both the in-distribution and out-of-distribution data, how is this achieved?**
>
> To ensure uniformity in the mapping, the output space of the M-class DNN classifier (M=10) used in the hash function was increased by subdividing it into (m=2) sub-classes yielding a total of M*m classes. We empirically found this to have enough dispersion for both in and out of distribution data to offer good security.
>
> [1] Mika Juuti, Sebastian Szyller, Samuel Marchal, and N Asokan. Prada: protecting against dnn model stealing attacks. In 2019 IEEE European Symposium on Security and Privacy (EuroS&P), pp. 512–527. IEEE, 2019.

---

### Official Review · AnonReviewer3 · 2020-10-28
**EDM's access to the adversary's OOD dataset**

**Rating:** 5
**Confidence:** 5

**Review:**

This paper tackles a timely problem of protecting deep neural networks from mode stealing attacks. This paper proposed an Ensemble of Diverse Model to provide diversity prediction for the adversary’s OOD query. The main contribution of this paper is the introduction of the diversity loss function on OOD data and the discontinuous prediction provided by ensembled models. The results show that EDM defense reduces the accuracy of stolen models.

Pros:
1.	The idea of using ensemble of diverse models to create discontinuous predictions on OOD datasets is interesting.
2.	The experimental evaluation is comprehensive. Five datasets and three attacks are investigated in the evaluation. Adaptive attacks using average predictions are considered in the evaluation as well.
3.	This paper is well-written. Figures make the paper easy to understand.

Cons:
1.	My major concern is the auxiliary OOD dataset used by EDM. Does the defender have access to the adversary’s OOD dataset? In the experiments, although the EDM used auxiliary OOD datasets, these auxiliary OOD datasets are more complex than the adversary’s OOD datasets. If the adversaries have access to the auxiliary OOD datasets or use other complex OOD datasets, will the defense still work?
2.	As mentioned in the paper, the hashing function requires transformation invariance, so that the adversaries cannot get the average prediction. However, can the DNN-based perceptual hashing function be generalized and applied to other types of transformation? For example, to break the transformation invariance, the adversary can use different algorithms to craft noises into the input data.
3.	From Table 3, it seems that the performance of EDM on JBDA and JDBA-TR is not very well. My understanding is that since JBDA and JDBA-TR do not reply on a fixed OOD dataset, EDM trained on auxiliary OOD dataset may be well generalized to these attacks’ OOD data.
4.	The query budget of the adversary is set as 50,000. A larger number of queries are used in the previous attacks. For example, in Juuti et al. 2019, 102,400 queries are used to steal models trained on the MNIST dataset. Does the small query budget used in the paper limit the performance of attacks?
5.	Minor comments:
a.	What is CS in equation (1) and (2)?
b.	“A perceptual hashing algorithm is used prevent adaptive attacks”
c.	“Our empirical evaluations shows that”

---

> ### Author Response · Authors · 2020-11-18
> **Response**
>
> We thank the reviewers for their valuable and positive feedback. We are happy that the reviewers found our problem to be “well-motivated” and our idea to be “novel”. We are also glad that the reviewers found our paper to be “well-written” and our experimentation to be “extensive”. We hope to address the concerns raised by the reviewers below:
>
> **Q1. If the adversaries have access to the auxiliary OOD datasets or use other complex OOD datasets, will the defense still work?**
>
> An adversary who uses the auxiliary OOD dataset to query EDM would likely obtain a worse clone accuracy compared to using other OOD datasets. For example, for the EDM CIFAR-10 model trained with the Tiny-Images auxiliary OOD dataset, using Tiny-Images to perform model stealing yields a clone accuracy of 34.95 (0.37x), which is lower than the clone accuracy of 51.34 (0.52x) obtained by using CIFAR-100. This is because the models in EDM are trained to produce maximally dissimilar (i.e. discontinuous) predictions on the auxiliary OOD dataset. Thus, an adversary who uses this same OOD dataset to query the target model obtains highly discontinuous predictions, resulting in poor clone accuracy.
>
> **Q2. As mentioned in the paper, the hashing function requires transformation invariance, so that the adversaries cannot get the average prediction. However, can the DNN-based perceptual hashing function be generalized and applied to other types of transformation? For example, to break the transformation invariance, the adversary can use different algorithms to craft noises into the input data.**
>
> If the adversary had white-box/black-box access to the predictions of the DNN used in our perceptual hashing algorithm, the adversary could easily pick a perturbation algorithm with a high likelihood of changing the output of the hash. However, since the DNN model used in the hash function is kept secret and the adversary does not have access to the output of the hash function, it is difficult for the adversary to  find a perturbation algorithm that can break the transformation invariance. Additionally, querying the target model with a set of transformed inputs opens up the attack to detection using methods such as PRADA (Juuti et al. 2019), which can limit the applicability of such adaptive attacks.
>
> **Q3. From Table 3, it seems that the performance of EDM on JBDA and JDBA-TR is not very well. My understanding is that since JBDA and JDBA-TR do not rely on a fixed OOD dataset, EDM trained on auxiliary OOD dataset may be well generalized to these attacks’ OOD data.**
>
> For most of the attacks and datasets, we found that the auxiliary OOD dataset does generalize well to the attacker’s OOD dataset. However, in some cases (FashionMNIST with JBDA) the benefits of EDM are less pronounced possibly due to the observation made by the reviewer. We acknowledge this in section 5.3--”With the JBDA and JBDA-TR attacks, the benefits of EDM are less pronounced as these attacks use perturbed versions of in-distribution examples to query the target, which may not produce large discontinuities in the predictions.”
>
> **Q4. A larger number of queries are used in the previous attacks. For example, in Juuti et al. 2019, 102,400 queries are used to steal models trained on the MNIST dataset. Does the small query budget used in the paper limit the performance of attacks?**
>
> We’ve added evaluations with 2x the query budget (100K instead of 50K) in Appendix A.6. For the JBDA and JBDA-TR, the clone accuracies plateaus before the query budget we use in our paper, therefore more trials do not provide significant benefits (this result is consistent with the observations made by Juuti et al. 2019). For KnockoffNets, the attacker is limited not only by the query budget but also by the amount of surrogate data to query the target model with. For the surrogate datasets where more data is available, we show that the benefits of our defense persist even if the query budget is doubled.
>
> **Q5 a.Minor comments: a. What is CS in equation (1) and (2)? b. “A perceptual hashing algorithm is used to prevent adaptive attacks” c. “Our empirical evaluations shows that”**
>
> CS is the Cosine Similarity. Thank you for these comments. We have updated the paper to fix these issues.

---

### Official Review · AnonReviewer2 · 2020-10-28
**Effective Model Defense Technique, Need More Experimental Evaluation**

**Rating:** 6
**Confidence:** 3

**Review:**

This paper proposes a simple but effective way to avoid model stealing. To the best of my knowledge, the idea is novel. The writing of this paper is quite clear and experimental results also show the effectiveness of proposed method. By introducing diversity loss, the model ensemble tends to produces discontinuous outputs which is hard to be distilled. I still have some questions:

1. In your experiments, there is an assumption that the thief knows what exactly the model archs. Given this prior knowledge, what will happen if the thief use a more complex/bigger model to distill ?

2. It seems  inconsistent results are observed on Flowers17 dataset, any explanation?

3. There is another assumption that the attackers are constrained by attach budget (50000 trials), what will happen if more trials or unlimited trials are allowed? I think this question is important because this is often true scenario if the attacked models are of high business value.

4. Have you tried your method on larger tasks such as ImageNet or other fields, for example machine translation and speech recognition?

---

> ### Author Response · Authors · 2020-11-18
> **Response**
>
> We thank the reviewers for their valuable and positive feedback. We are happy that the reviewers found our problem to be “well-motivated” and our idea to be “novel”. We are also glad that the reviewers found our paper to be “well-written” and our experimentation to be “extensive”. We hope to address the concerns raised by the reviewers below:
>
> **Q1. What happens if the surrogate model is more complex?**
>
> We’ve updated our paper with results from using a more complex surrogate model architecture in Appendix A.5. We obtain similar clone accuracies as before with a more complex model.
>
> **Q2. It seems inconsistent results are observed on the Flowers17 dataset, any explanation?**
>
> We’ve fixed a typo in our results (Flowers-17, hard label, JBDA-TR, EDM). Please let us know if you have a specific concern with the results aside from this.
>
> **Q3.  What will happen if more trials or unlimited trials are allowed?**
>
> We’ve added evaluations with double the query budget (100K instead of 50K) in Appendix A.6. For the JBDA and JBDA-TR, the clone accuracies plateau before the query budget we use in our paper, therefore more trials do not provide significant benefit (this result is consistent with the observations made by Juuti et al. 2019 [1]). For KnockoffNets, the attacker is limited not only by the query budget but also by the amount of surrogate data to query the target model with -- more queries than the amount of surrogate data do not help. For the surrogate datasets where more data is available, we show that the benefits of our defense persist even if the query budget is doubled.
>
> **Q4. Have you tried your method on larger tasks such as ImageNet or other fields, for example, machine translation and speech recognition?**
>
> We haven’t evaluated our defense on ImageNet. To the best of our knowledge, model stealing attacks have not been demonstrated to work on the Imagenet task in the data-limited setting, so it may be premature to evaluate defenses. However, we do evaluate our defense on 5 different datasets with resolutions ranging from 28x28 (FashionMNIST) to 256x256 ( Flowers17) and number of classes ranging from 10 (FashionMNIST) to 100 (CIFAR-100). Most of the MS attacks focus on stealing models in the computer vision domain. While we haven’t tried our method on other fields like machine translation and speech recognition, it would be interesting to explore these attacks in other domains as a future direction.
>
> [1] Mika Juuti, Sebastian Szyller, Samuel Marchal, and N Asokan. Prada: protecting against dnn model stealing attacks. In 2019 IEEE European Symposium on Security and Privacy (EuroS&P), pp. 512–527. IEEE, 2019.

---

### Decision · Program_Chairs · 2021-01-07
**Final Decision**

**Decision:**

Accept (Poster)

**Comment:**

This paper proposed an ensemble of diverse models as a mechanism to protect models from theft.
The idea is quite novel. There are some concerns regarding the robustness of the hashing function (that I share), however not every paper has to be perfect, especially when it introduces a novel setup.

AC